# Micro-Texture Analyses of a Cold-Work Tool Steel for Additive Manufacturing

**DOI:** 10.3390/ma13030788

**Published:** 2020-02-09

**Authors:** Jun-Yun Kang, Jaecheol Yun, Byunghwan Kim, Jungho Choe, Sangsun Yang, Seong-Jun Park, Ji-Hun Yu, Yong-Jin Kim

**Affiliations:** Korea Institute of Materials Science, 797 Changwon-daero, Changwon, Gyeongnam 51508, Koreaa921101@kims.re.kr (B.K.); drgb0443@kims.re.kr (J.C.); nanoyang@kims.re.kr (S.Y.); hyega@kims.re.kr (S.-J.P.); jhyu01@kims.re.kr (J.-H.Y.); yjkim@kims.re.kr (Y.-J.K.)

**Keywords:** additive manufacturing, selective laser melting, tool steel, microstructure, texture, electron backscatter diffraction

## Abstract

Small objects of an alloy tool steel were built by selective laser melting at different scan speeds, and their microstructures were analyzed using electron backscatter diffraction (EBSD). To present an explicit correlation with the local thermal cycles in the objects, prior austenite grains were reconstructed using the EBSD mapping data. Extensive growth of austenitic grains after solidification could be detected by the disagreement between the networks of carbides and austenite grain boundaries. A rapid laser scan at 2000 mm/s led to less growth, but retained a larger amount of austenite than a slow one at 50 mm/s. The rapid scan also exhibited definite evolution of Goss-type textures in austenite, which could be attributed to the growth of austenitic grains under a steep temperature gradient. The local variations in the microstructures and the textures enabled us to speculate the locally different thermal cycles determined by the different process conditions, that is, scan speeds.

## 1. Introduction

Recently, additive manufacturing (AM) has been intensively investigated and has found practical usages in many sections of manufacturing industries [1,2,3]. Various AM technologies have been developed for various materials [2,3,4,5]. For metallic materials, it is generally implemented by selective melting of powder with laser or electron beams [3,4,5,6,7,8,9,10]. Many kinds of metallic powder have been tested for application in mechanical or functional parts [3,6,7,8,9,10,11,12,13,14,15,16,17,18,19,20,21,22] and a few new alloys dedicated to AM are being developed [14,21,22].

As they constitute the major part of the current engineering metals, steels can make a great contribution to the spread of AM applications. Among various steels, tool steels are well suited to AM [17,18,19,20,21,22,23], and become parts of the representative commercial steel powder for AM [23]. They generally have high alloy contents and suffer from excessive heterogeneities owing to segregation when they are provided via the conventional ingot metallurgy route [24]. Thus, the fine and homogeneous microstructures produced by selective melting of powder [17,18,19,20,21,22] should be much more attractive. Machinability of the conventional wrought or cast tool steels is an important requirement by toolers [24], and the near-net-shape by AM should be another attraction to these difficult-to-machine materials. The conventional powder metallurgy routes have been adopted to some commercial alloy tool steels to attain the microstructural refinement and homogeneity [24]. However, the sizes and shapes of the products by their consolidation processes are often restricted, and they would require a long process time to make the final parts of tools and dies from powder [24]. On the other hand, AM by the selective melting has better flexibility and promptness to meet various demands (such as geometry and lead time) by various consumers [1,2], and thus is more suited to the nature of the tools and dies market, that is, small quantity and customization. For other metallic materials, AM is preferred in these types of markets such as the metallic implants of titanium alloys [23].

In spite of the aforementioned advantages, only a limited number of documents are still found on the AM applications of tool steels [17,18,19,20,21,22], while most of them dealt with standard hot-work or plastic tool steels such as AISI H13 and P20 [17,18,19,20]. It would be harder to find ones that dealt with more highly alloyed cold-work tool steels [21,22]. Although those works reported some results of microscopic observations [17,18,19,20,21,22], more detailed and quantitative information on the morphologic and the crystallographic characteristics of the microstructures would improve the current knowledge for the AM of tool steels.

The mostly martensitic matrix of tool steels generally causes difficulties in the characterization of their microstructures. Owing to the complicated subdivision of the parent phase (i.e., austenite) by martensitic transformation [25], it would be substantially difficult to deduce a simple correlation between microstructural features (e.g., grain size and texture) and process conditions.

In this study, quantitative micro-texture analyses using electron backscatter diffraction (EBSD) were performed on the powder and the AM objects of a cold-work tool steel. In order to gain an explicit correlation between the microstructures and process conditions, a recent technique for the reconstruction of prior austenite grains [26,27,28] was utilized. Thus, this study could provide a useful example of the procedures to characterize the microstructures of tool and other martensitic steels for AM.

## 2. Material and Methods

The chemical composition of the steel powder is presented in Table 1. This alloy was originally designed as a wrought tool steel for cold stamping applications [29]. It exhibited an excellent balance between wear resistance and toughness, but had problems with segregation and machinability [29]. Thus, this non-commercial alloy was chosen as a candidate to test the feasibility of high-alloyed tool steels for AM. Some preliminary characterization of the microstructures in the powder and the AM objects can be found in an authors’ previous work [22].

The powder was fabricated with a high-pressure hot gas atomizer, HERMIGA 100/25 by PSI [22]. With the powder of diameters 10–45 µm, small cuboidal objects were built by selective laser melting (SLM) using a powder bed fusion (PBF) type printer, ConceptLaser by MLab. The power and diameter of the laser beam were 90 W and 110 µm, respectively. The thickness per single layer of deposition was 25 µm and the spacing between the scan line was 80 µm. The scan direction was repeatedly inversed between the adjacent scan lines. The substrates were plates of austenitic stainless steel, AISI 304. In Figure 1, the schematic illustrations of the scan pattern and the designed objects are presented with dimensions. Two AM objects built with a scan speed of 50 and 2000 mm/s (designated as C50 and C2k, respectively) were used for the micro-textural characterization.

Metallographic sample preparation was performed to examine the powder and the AM objects with EBSD. A small amount of the powder was mounted in conductive resin, mechanically ground, and polished. It was finally chemo-mechanically polished with colloidal silica. The AM objects were sectioned to examine the surfaces that contained the build and the scan directions (BD and SD, respectively), as shown in Figure 1. These surfaces were mechanically ground, polished, and finally electro-polished with LectroPol-5 by Struers. The electrolyte was 10% perchloric acid in ethanol at −20 °C.

For the micro-texture analyses, an EBSD system, which was installed in a field emission scanning electron microscope, JSM 7001F by JEOL (Tokyo, Japan), was used. It consisted of NordlysNano detector and AZTEC software by Oxford Instruments. The orientation mapping was performed with step sizes of 0.05–0.5 µm, in which the beam energy and the probe current were 15 kV and 5 nA, respectively. As shown in Figure 1, the different sampled regions were examined for the AM objects. The regions of the upper and the lower layers were the sub-surface regions, about 100 µm apart from the top and the bottom surfaces, respectively, owing to the rounded edges of the specimens by electro-polishing.

For postprocessing of the EBSD mapping data, an open source MATLAB^TM^ toolbox, MTEX [30,31,32], was utilized. As the final martensitic transformation hid much information about the microstructure evolution during the processing of martensitic steels, the reconstruction of prior austenite grains was performed using the MATLAB tool by Nyyssönen [27,32], which was based on the methods by Gomes et al. [28] and MTEX. In the reconstruction procedure, during which the clustering [33,34] of martensitic sub-structures was performed, single-element clusters were prevented from reconstruction as they would make noises in the results [26]. The other settings of parameters followed the recommended ones, that is, 3° of misorientation angle for the detection of martensitic subunits and 1.6 of inflation power for clustering [27,28,32]. After the reconstruction, grains that were too small, which contained less than five measure points (pixels), were rejected in the calculation of grain statistics. In the austenitic grain detection, the definition angle for grain boundary misorientation was 5° and the twin boundaries defined by the 60°//<111> misorientation (within 5° tolerance) were rejected.

## 3. Results

The microstructures of the powder revealed by the micro-texture analyses are presented in Figure 2. A bright network by very fine carbides was observed among the dark matrix in the forward scattered electron image of Figure 2a. In the authors’ previous work [22], it was revealed that the carbide network formed during solidification and consisted of Nb-rich MC and Mo-rich M_2_C. Figure 2b is the phase map from the orientation mapping on the area of Figure 2a. Owing to the very fine microstructures of the powder, the finest step size, that is, 0.05 µm, was set. The matrix was mostly indexed as body-centered cubic (BCC) steel represented by green color, and thus could be regarded as martensite. Although the carbides of red (MC) or magenta (M_2_C) colors are vaguely revealed in Figure 2b owing to their fineness, it is clear that a substantial amount (16.7%) of face-centered cubic (FCC) steel (i.e., austenite) with blue color was retained along the carbide network from the comparison with Figure 2a. The fine carbides and the martensitic subunits of high dislocation contents would make more overlaps or noises of diffraction patterns, and thus a considerable amount (22.3%) of data points could not be indexed, and constitute the vacant (white) pixels in Figure 2b. Figure 2c,d are the inverse pole figure maps (IPF) that represent the crystal directions of BCC and FCC steels parallel to the normal of the observation surface from the raw and the reconstructed data, respectively. In Figure 2d, the grain and twin boundaries of the reconstructed austenite are drawn with thick black and thin gray lines, respectively, from which the average grain size (Feret diameter) of 3.27 µm can be obtained. The grain boundaries were detected by the misorientation angles larger than 5°, while the twin boundaries were defined by the misorientations of 60°//<111> within 5° tolerance. A considerable coincidence can be found between the carbide network in Figure 2a and the grain boundary network of the reconstructed prior austenite in Figure 2d. This indicates that the carbide network projects the as-solidified structures of austenitic grains. On the other hand, the martensitic structure by the raw data, that is, Figure 2c, hardly gives this kind of information.

Figure 3 and Figure 4 show the representative microstructures at the mid-layers of the AM objects by orientation mapping with a fine step size (0.1 µm). The carbides were disregarded in the mapping with step sizes >0.05 µm because of their small sizes and negligible influence on phase maps. The characteristic carbide networks were also found in Figure 3a and Figure 4a, while the network spacing in C2k was the finest. In Figure 3b, C50 exhibited the preferential distribution of retained austenite along the carbide network as in the powder, whereas the fraction (11.4%) was greatly reduced. On the other hand, in Figure 4b, it would be hard to find a definite tendency of retained austenite site along the network, and the fraction was greatly increased (22.5%) in C2k. As shown in Figure 3d and Figure 4d, the critical difference of the AM objects from the powder was the total loss of the coincidence between the carbide network and the austenite boundaries. The prior austenite grains were much larger than the cells by the carbide network and exhibited an elongated shape along BD. Meanwhile, the sizes of the austenitic grains were finer in C2k than in C50. Additionally, this information is hardly recognizable in Figure 3c and Figure 4c by the raw data.

Figure 5 shows the representative microstructures of the AM objects according to the different sampled regions. These were obtained by orientation mapping with the largest step size (0.5 µm) to collect enough numbers of reconstructed austenitic grains. A total of 3–4 maps per each region were obtained to collect more than 1600 grains per each region for statistical relevance. The AM objects exhibited distinctly different characteristics according to the process condition, that is, the scan speed, and to the regions in them.

The phase maps in Figure 5 clearly show the tendency to retain more austenite in C2k. Figure 6 presents the average fractions over larger areas and confirms this. For all the sampled regions, the retained austenitic fraction in C2k was 22.5%, while that in C50 was 11.4%. In addition, C50 exhibited more dependency on the sampled region, which is clear in Figure 6. More austenite (blue pixels) was present as approaching the top surface of C50, which was also clearly visible in Figure 5a.

From the reconstructed austenitic orientation maps in Figure 5, the evolution of grain sizes prior to the final martensitic transformation can be addressed according to the scan speed and the sampled region. C50 definitely had larger grains than C2k and the grain sizes also had a considerable correlation with the sampled regions in both specimens. Figure 7 presents the average grain sizes according to the scan speed and the regions using the reconstructed maps such as Figure 5c–d,g–h,k–l. Figure 7a corresponds to the average values over all the detected grains, but fails to represent the characteristics visible in Figure 5 because of very fine grains within relatively coarse grains. Figure 7b,c present the average values after filtering small grains of diameters less than 5 and 10 µm, respectively. They agree much better with the reconstructed maps in Figure 5 than in Figure 7a. Qualitatively, Figure 7b would best describe the grain size evolution presented in the reconstructed maps. In Figure 7b, the larger austenitic grains in C50 are again evident irrespective of the sampled regions. In C50, the average sizes increased from the lower to the upper layers continually. On the other hand, in C2k, the grain sizes on the mid-layers were coarser than on the others.

The crystallographic textures of the austenite grains according to the scan speed and the regions are presented with the orientation distribution functions (ODFs) in Figure 8. C50 exhibited nearly random distributions of orientations in Figure 8a–d, whereas C2k had moderate to strong textures in Figure 8e–h. The major texture component was the orientation of <011>//BD and <100>//SD, that is, {011}<100>. The representation of this component was renowned as Goss orientations in rolled and annealed steels [35,36]. The intensity of texture in C2k varied according to the regions, with the strongest on the mid-layers.

## 4. Discussion

The different microstructures and textures listed above are the consequences of the different thermal cycles depending on the process conditions and the sampled regions. Therefore, in this work, the local variations of the thermal cycles could be speculated from those of the microstructures and textures.

### 4.1. Carbide Networks and the Reconstructed Austenite Grain Structures

The Nb-rich MC carbides in alloy tool steels are highly stable against additional thermal cycles after they form [37]. Thus, the carbide network in this study should be a good trace marker for the initial grain structures by solidification [22]. The single heating (i.e., melting) and cooling cycle in the fabrication of the powder must induce single austenite-to-martensite transformation and the fine size of the powder guaranteed a rapid cooling. In this circumstance, the very initial network of the austenitic grain boundaries from solidification should be well preserved and coincide with the carbide network, as shown in Figure 2. On the other hand, in Figure 3 and Figure 4, the two networks never coincided. This indicates that the grain structures by solidification were destroyed by the subsequent growth of the austenitic grains. This post-solidification growth in SLM could be hardly detected, and thus has not been reported in a martensitic steel. In this study, the reconstruction of austenite grains and the stable Nb-rich carbides could explicitly reveal its existence.

This growth would occur during the cooling directly after solidification depending on cooling rates. Meanwhile, the more closely spaced network in Figure 4a than in Figure 2a indicates that a more rapid solidification and cooling was possible in C2k than in the powder. However, this contradicts the much greater growth in C2k than in the powder, which should be attributed to the repeated cycles of the growth in the AM objects via the repeated heating.

### 4.2. Retained Austenite Fraction and Thermal Cycles in the Parts of the AM Objects

Concerning only the solidification, a rapid solidification and large undercooling should lead to the solidification of austenite with a high C content [38], which in turn gives a tendency to retain more austenites [39]. This might partly account for the much more retained austenite in C2k than in C50, as shown in Figure 3, Figure 4, Figure 5 and Figure 6. However, as there were additional heating and cooling cycles by the deposition of successive layers in the AM objects, the characteristics of austenite prior to the final martensitic transformation should deviate from the very initial ones by solidification. The different energy input by the different scan speed has a major influence on these thermal cycles [10,11]. In C50, a higher density of energy should lead to a greater build-up of temperature, and thus result in prolonged thermal cycles with slower cooling. More chances should be given to the partitioning of C from austenite to carbides, which would retain less austenite in the final microstructures.

In Figure 6, both objects had the least amount of retained austenite on their mid-layers, which should reflect the more prolonged thermal cycles than in the other regions. On the lower layers, a fast cooling by an effective heat sink, that is, the metallic substrate, should restrict the effect of thermal cycles. On the upper layers, the effect should be also restricted as only a few additional layers were deposited. Additionally, cooling could be enhanced from the top surface after the end of the process. In Figure 6, C2k exhibited much less dependency on the sampled regions, which reflected the limited build-up of temperature owing to a low density of energy.

Additionally, from Figure 2b, Figure 4b, Figure 5b,f,j, and Figure 6, it was known that less austenite was retained in the powder than in C2k. As already noticed from the finer carbide network, this AM object should experience faster solidification and cooling, and thus retained more austenite than the powder in spite of the repeated thermal cycles.

### 4.3. Austenitic Grain Sizes and Thermal Cycles in the Parts of the AM Objects

Although the prior austenite grain boundaries are not obvious in the final martensitic microstructures, it is generally accepted that the prior austenitic grain size contributes significantly to mechanical properties [40]. Additionally, in the correlation with process conditions at high temperatures, it can be more informative than the sizes of the martensitic subunits (such as lath, plate, packet, and block). Thus, the grain size analysis on the austenitic grains prior to the final transformation would be more effective for the AM of martensitic steels.

Owing to the exceptionally fine grains in the reconstructed maps of C50, that is, Figure 3d and Figure 5c,g,k, the statistics in Figure 7a do not accord with those in the reconstructed maps. It is especially evident in Figure 3b,d that these fine grains within the coarse reconstructed grains are mostly the fine retained austenites adjacent to the carbide network. The different orientations of the large reconstructed austenites from those of the fine retained ones might simply indicate errors in the reconstruction procedure. However, there could also be other possibilities. (1) The modification of martensitic orientations was possible during thermal cycles, and thus can increase the deviation of the reconstructed orientations from the retained ones. (2) If there were repeated cycles of re-austenitization and martensitic transformation with parts of the initially retained austenite conserved, the finally reconstructed orientations could be substantially different from the retained ones. (3) Some retained austenites in the phase maps could also correspond to the mis-indexed data from coarse MC carbides because of the similar crystal structures. All of these possibilities should be more probable in C50, which experienced slower solidification and more extended thermal cycles, and included much more of the fine intra-granular grains in the reconstructed maps. Therefore, these types of the very fine austenitic grains were not simple artifacts from the numerical procedure of the reconstruction, although they were filtered out in Figure 7b,c to follow the major trend of the grain growth in Figure 5c,d,g,h,k,l.

In Figure 7b,c, the smaller average sizes in C2k indicate a rapid cooling in an austenitic regime, which is consistent with the previous discussion on the retained austenite fraction. The finest sizes at the lower layers of both objects well reflects the rapid cooling due to the substrate, and the increased sizes at the mid-layers can be explained by slowed cooling due to temperature build-up in extended thermal cycles. As there were only a few layers of additional deposition on the upper layers, the limited thermal cycles and the cooling from the top surface should limit the grain growth. However, the average size continued to increase toward the upper layers in C50. Thus, there should be a continued increase in peak temperature in C50, which even outweighed the effect of the limited thermal cycles and the enhanced cooling on the upper layers. The increased peak temperature should also result in the increased fraction of retained austenite [24] on the upper layers, as shown in Figure 6. On the other hand, in C2k, the cooling driven by the top surface should be more influential, which resulted in less grain growth on the upper layers.

### 4.4. Austenitic Textures and Thermal Cycles in the Parts of the AM Objects

As the austenitic textures in Figure 8 were actually swept out by the final transformation, these would have little influence on the properties of the AM objects, such as wear resistance. However, they can be also a useful indicator of the process conditions.

Solidification microstructures often have morphological and crystallographic textures along temperature gradient, which depends on the process conditions [8,10,16,39,41]. In the solidification of cubic metals, the columnar shape of grains whose principal axes are <100> directions along the maximum temperature gradient is often reported [8,10,39,41,42]. Thus, a BD//<100> texture was generally mentioned in the SLM of cubic metals [8,10,41,42].

On the other hand, the carbide networks in Figure 2, Figure 3 and Figure 4 indicate that the initially solidified austenitic grains in the AM objects as well as the powder should have mostly equiaxed morphologies and hardly have textures by rapid solidification [10]. Nevertheless, the reconstructed austenite grains in C2k exhibited the textures shown in Figure 8e–h. The Goss-type texture was the strongest on the mid-layers where the grain growth was also the most extensive, as shown in Figure 5 and Figure 7. Thus, in this study, it is more reasonable to attribute the austenitic textures to the grain growth after solidification. The definite textures in C2k imply that the textures by the post-solidification growth should depend on the steepness of temperature gradient like solidification textures [10]. The probable destruction of solidification textures by repeated transformations could be also considered. However, this cannot account for the weakened texture on the upper layers of C2k, where the repeated transformations should be less probable than on the mid-layers.

The characteristic Goss component in C2k instead of the generally expected BD//<100> could also support the possible role of the post-solidification growth under a temperature gradient. Interestingly, a few papers in the literature [43,44] also reported similar BD//<110> textures in the AM objects of an austenitic steel (AISI 316L). However, in spite of the above discussions, the current results are insufficient to justify the development of the specific texture component. A more systematic and intensive work is currently planned and the results will be provided in the near future.

## 5. Conclusions

In this study, micro-texture analyses on the powder and the AM objects of an alloy tool steel were performed. The resulting micro-textural data could provide useful information about the evolution of microstructures in the AM objects, their correlation with the local thermal cycles, and the process conditions, which are summarized below.

(1) The carbide networks were inconsistent with the austenitic grain structures in the AM objects, from which extensive post-solidification growth of austenitic grains during the SLM process could be identified.

(2) High density of energy input by a slow laser scan resulted in smaller fractions of retained austenite and larger grain sizes because of more intensive temperature build-up, and subsequently extended thermal cycles.

(3) There were local variations of microstructures that originated from the variations of local thermal cycles by the different scan speeds. Enhanced cooling near the metallic substrate resulted in large fractions of retained austenite and fine grain sizes. On the mid-layers, retained austenitic fractions decreased with more extensive grain growth owing to the extended thermal cycles by temperature build-up. Limited deposition of energy and shortened thermal cycles near the top layers increased retained austenitic fractions, whereas the extent of austenitic grain growth could vary depending on the temperature built up and the cooling by the top surface.

(4) The evolution of the Goss-type textures exhibited a close correlation with the post-solidification growth. Stronger textures were developed with the austenitic grain growth under steeper temperature gradient by a fast laser scan.

The information on the microstructural variations according to the scan speeds (50 and 2000 mm/s) could present a useful guidance to manufacture an object for real-world application. Besides the avoidance of the reported porosity by over-melting [22], the more closely spaced carbide network and the more homogeneous distribution of the retained austenite at rapid scan should be more desirable for the resistance against abrasion and fatigue [24] after an adequate tempering heat treatment. However, care must be given to avoid excessive residual stress due to a steep temperature gradient or incomplete melting of powder [22].

## Figures and Tables

**Figure 1 materials-13-00788-f001:**
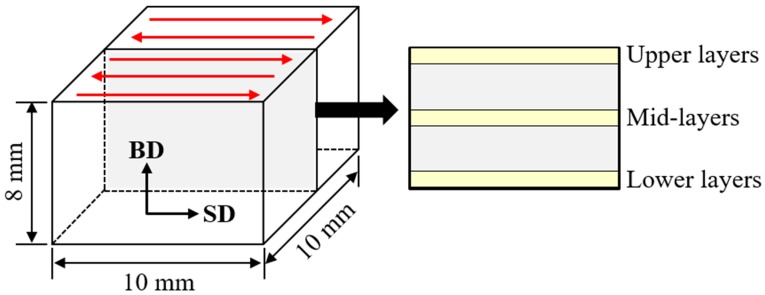
Additive manufacturing (AM) object by the selective laser melting process (SD: scan direction, BD: build direction, the scan pattern is indicated with the alternate arrows) [22] and its cross-sectional specimen for the microanalyses with the indicated sampled regions.

**Figure 2 materials-13-00788-f002:**
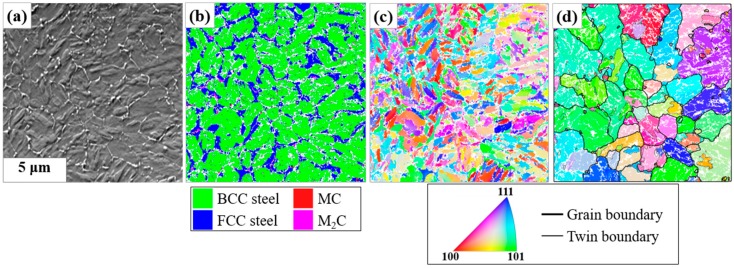
Microstructure of the powder: (**a**) image from forward scatter detectors (FSDs); and (**b**) phase map by the different phase colors, (**c**) original and (**d**) reconstructed inverse pole figure (IPF) map referred to the normal of the observation surface (Refer to the given color key for a cubic crystal. Grain boundaries were drawn for misorientation angle >5° and the twin boundaries were defined by the misorientaions of 60°//<111> within 5° tolerance).

**Figure 3 materials-13-00788-f003:**
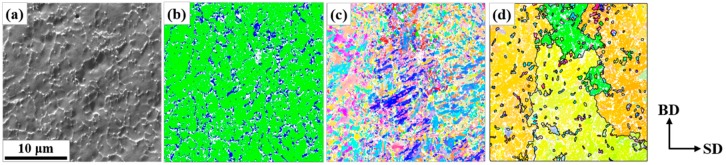
Representative microstructure on the mid-layers of the AM object built at 50 mm/s (C50): (**a**) FSD image, (**b**) phase map, and (**c**) original and (**d**) reconstructed IPF map referred to BD (The phase colors, the IPF color key, and the convention for boundary drawing followed those in Figure 2. The carbides in (**b**) and the twin boundaries in (**d**) were removed for cleanliness).

**Figure 4 materials-13-00788-f004:**
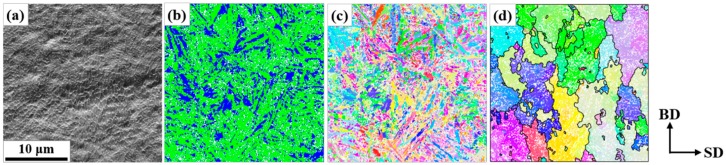
Representative microstructure on the mid-layers of the AM object built at 2000 mm/s (C2k): (**a**) FSD image, (**b**) phase map, and (**c**) original and (**d**) reconstructed BD-IPF map (The phase colors, the IPF color key, and the convention for boundary drawing followed those in Figure 2. The carbides in (**b**) and the twin boundaries in (**d**) were removed for cleanliness).

**Figure 5 materials-13-00788-f005:**
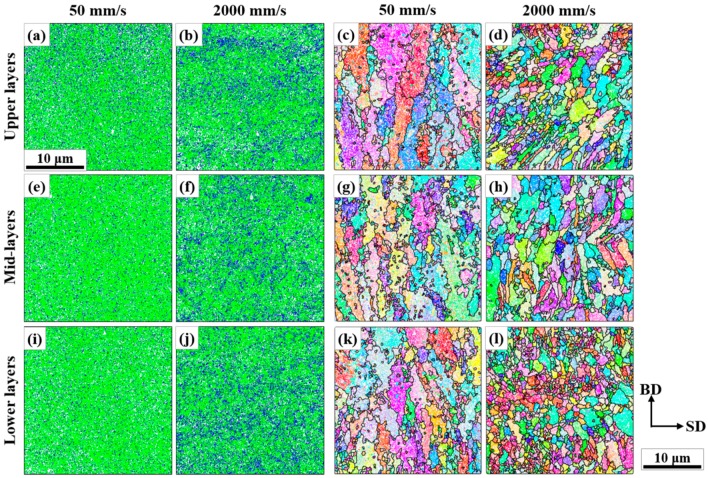
Representative microstructures on the different sampled regions in the AM objects: (**a**,**b**), (**e**,**f**), and (**i**,**j**) phase maps; (**c**,**d**), (**g**,**h**), and (**k**,**l**) reconstructed BD-IPF maps (Refer to the left and the upper parts of the figure for the information on the regions and the process conditions. The phase colors, the IPF color key, and the convention for boundary drawing followed those in Figure 2. The carbides in (**b**) and the twin boundaries in (**d**) were removed for cleanliness).

**Figure 6 materials-13-00788-f006:**
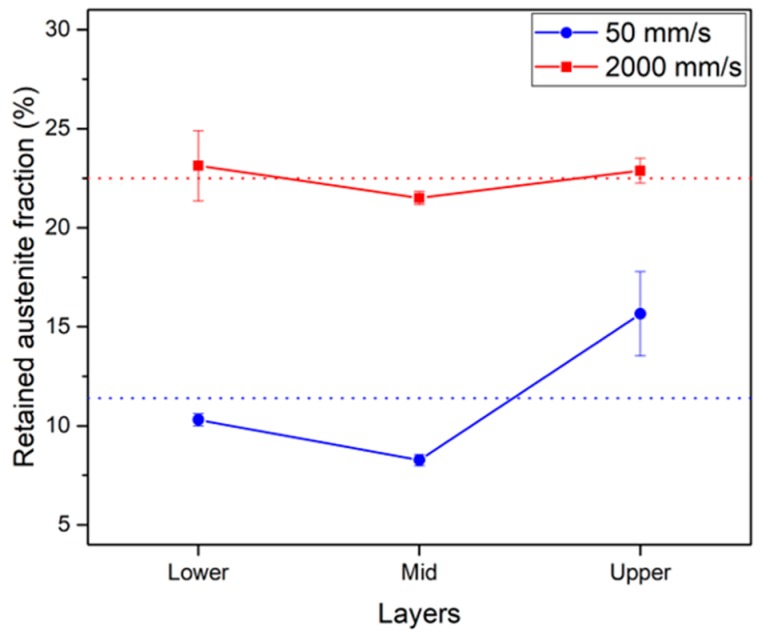
Average fraction of retained austenite according to the regions and the scan speeds of the AM objects (the error bars and the broken horizontal lines represent the standard errors and the average values for the whole sampled regions, respectively).

**Figure 7 materials-13-00788-f007:**
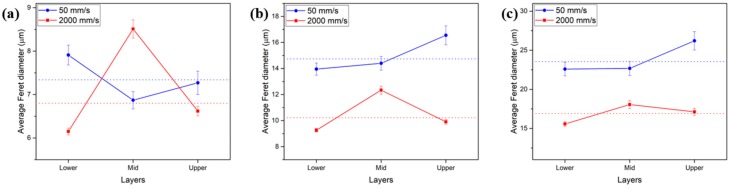
Average sizes of the prior austenite grains according to the regions and the scan speeds of the AM objects: (**a**) for all the detected grains, and (**b**) the grains of diameter >5 and (**c**) >10 μm only (the error bars and the broken horizontal lines represent the standard errors and the average values for the whole sampled regions, respectively).

**Figure 8 materials-13-00788-f008:**
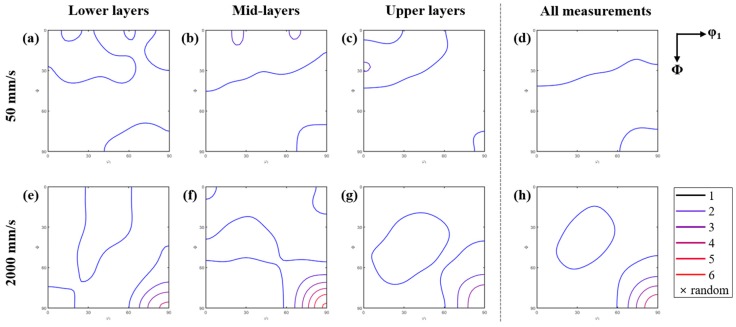
Orientation distribution functions (ODFs) on the φ_2_ = 45° section of the Euler space (Bunge’s convention) according to the regions and the scan speeds of the AM objects (Refer to the left and the upper parts of the figure for the information on the regions and the speeds. The sample reference axes 1 and 3 correspond to SD and BD, respectively): (**a**–**d**) ODFs of the lower, the mid-, the upper layers and all the sampled regions of C50 respectively, (**e**–**h**) ODFs of the lower, the mid-, the upper layers and all the sampled regions of C2k respectively

**Table 1 materials-13-00788-t001:** Chemical composition of the tool steel powder (wt. %) [22].

C	Si	Mn	Cr	Ni	Mo	Nb	V	W	Cu	Fe
0.89	0.20	0.30	5.29	0.17	2.78	1.22	0.42	0.19	0.19	Bal.

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
