# Peer review of "Micro-Texture Analyses of a Cold-Work Tool Steel for Additive Manufacturing"

_materials, 2020, doi:10.3390/ma13030788_

Round 1
Reviewer 1 Report
Comments to the manuscript “materials-692125”:
The authors attempted to investigate the microstructure distribution of the part prepared by selective laser melting. Samples from different regions of the part were examined by electron backscatter diffraction (EBSD). It was found that prior austenite grains were reconstructed owing to the local thermal cycles. Different laser scan rates were chosen to study the effect on the microstructure distribution. However, the description of the microstructure is not clear at all. The demonstration of the experimental results should be as clear as possible. Overall, the manuscript needs to be reorganized and the writing needs to be enhanced so as to improve the readability. The following suggestions should be taken into account before its resubmission.
>> The authors need to clarify how to read the different phases from the present phase mapping. For example, how can we learn from the color bar given in the top-left panel in Figure 2? Also how do we distinguish the grain boundary and twin boundary in the reconstructed map (or polarized map)?
>> It is terrible to display the phase map and the polarized image together. How can we correlate the different phases in the phase map and the polarized images? It makes confusion about the grain distribution in the two type of images.
>> Figure 2 shows the microstructure of the powder. What is the grain size in the powder? How to read the phase map in Figure 2b? How do we learn from Figure 2d about the grain boundary and twin boundary?
>> How to compare Figure 2 and Figure 5? Grain size? Phase change? Where are the metal carbides?
>> A scale bar is needed in Figure 1 and Figure 5.
>> The writing needs to be improved.
Author Response
We greatly appreciate your careful review and the valuable comments which guided a better way for our revised manuscript. Here, we give our answers to your comments point-by-point. Please note that the corresponding revisions are highlighted with green color in our revised manuscript.
(1) Overall, you suggested clearer demonstration of the experimental results and to improve readability. We agreed with your opinion totally, so carried out overall correction of expressions. Although not highlighted, you will find our sentences were substantially rewritten. Especially, we put more focus on the result section in order to describe the experimental results in much more detail. The important revisions are highlighted.
(2) We thought that you requested a detailed explanation to read the phase map and to distinguish the grain and the twin boundaries in the reconstructed map in Fig. 2. You can find those in the following location of the revised manuscript (the part is highlighted)
> page 7, lines 6-22
> caption of Fig. 2
We also modified the legends in Fig.2 which are commonly used through Figs. 2-5. Accordingly, the captions in Figs. 3-5 were also revised.
(3) You mentioned the ineffectiveness in the display of the phase map and the polarized image (we thought that you pointed the FSD images) together. You will find the redrawn phase maps in the following figures.
> Figs. 2-4
(4) You asked the grain size in the powder and again requested better explanation of Fig. 2. You can find our answer in the above (2). Specifically, the grain size can be found in the following location (highlighted).
> page 7, lines 18-20
(5) You requested the way to compare Fig. 2 and 5 concerning grain size and phase change, and wanted to know where the carbides were. Essentially, the two contain the same kind of information. However, Fig. 5 was intended for statistical analyses, thus in poorer resolution, but included much larger area of observation via using 10 times coarser step size. Thus, in Fig. 5, we disregarded the carbide phases which were too fine for mapping. In Fig. 2 by much finer step size, the two carbides are only vaguely visible along with the blue austenitic regions (they were only visible mostly as single pixels like noises when step size exceeded 0.05 microns). Due to their small volume fractions (< 2 %), this omission did not make a serious influence on the resulting phase maps. The comparison of grain size and phase constitution were naturally added in the course of the revision corresponding to your previous comments. You can find them in the following locations.
> page 7, lines 9-12, lines 18-20
> page 8, lines 5-7, lines 19-21
> page 9, lines 2-3, lines 3-6
(6) You suggested the addition of scale bars in Fig. 1 and 5. Actually, the scale bars were evidently presented in the original Fig. 5. In the original Fig. 1, the dimensions were also presented. Currently, we could not know the reason why the scale bars and the dimensions were not visible to you. Anyways, we made a few revisions in the following locations.
> page 5, lines 13-14
> Fig. 5, an additional scale bar was given at the bottom right.
(7) You requested the improvement of writing. As we already answered in (1), we substantially revised our expressions overall. We totally agreed that our original manuscript had poor readability, thus did our best for readers’ understanding. Thank you very much for all of your comments.
The above answers are also provided via an attached file (Reviewer1.docx).

Reviewer 2 Report
Referee Report
on paper
“Micro-texture Analyses of a Cold-Work Tool Steel for Additive Manufacturing”
by authors Jun-Yun Kang, Jaecheol Yun, Byunghwan Kim, Jungho Choe, Sangsun Yang, Seong-Jun Park, Ji-Hun Yu1 and Yong-Jin Kim
submitted to Materials
The paper focuses on the results of the micro-texture investigations of the powder and the samples from an alloy tool steel. The samples were fabricated via well-known technique – the additive manufacturing. The presented results can cause great interest among readers of the Journal, because they relate to the current manufacturing methodology and widely used material. The authors showed a difference in the structure of steel samples that were made at different scanning speeds of the laser beam and at different temperature conditions. Perhaps the results and conclusions will be useful for practical use in production, as well as for a deeper fundamental understanding of the processes of formation of bulk materials from powders in the process of additive manufacturing under the influence of temperature. Despite this, I would like to make a few comments. I think the paper needs major revision and after second consideration of the paper with adjustments, it may be published in Materials
Comment 1
The introduction is almost entirely devoted to the advantages and features of additive manufacturing. There is no justification for the material choice. There is no doubt that the common steel will be a good candidate for use as a constructional material for additive manufacturing. However, at least a comparison with other materials usual for the method should be made in the introduction. After all, the journal is called Materials.
Comment 2
Very slight changes in the microstructure were noted in the article especially between upper, mid and lower layers. For example, in the fig 9 the average feret diameter varied between + - a few μm. What is the reason for such accuracy? Doubts arise for the reason that this type of steel is quite platichny and in the process of cutting, grinding and polishing, significant changes in the microstructure may appear.
Comment 3
Based on comment 1, I recommend adding the error bars to the graphs in Figures 6 and 9.
Comment 4
The main place in the paper is rightly given to the change in grain size depending on the conditions for the samples manufacture. In Figs. 7 and 8, the grain size (feret diameter) distributions according to the scan speed and the regions are presented. Graphs in Fig. 7 are the dependences of number of fraction on diameter. At the same time, Fig. 8 is the dependence of the fraction area on the ferret diameter. By and large, this is the same information presented in a different form. Authors should explain the need for data duplication or leave only one type of distribution.
Comment 5
As a rule, classical works in the field of materials science include a description of the chemical composition, structure, and properties of the materials. You can see it in the fallowing paper for example https://doi.org/10.1007/s11665-018-3483-7, https://doi.org/10.1016/j.vacuum.2018.07.017 and doi:10.1088/1757-899X/256/1/012022. Of course, this is not a requirement for each paper. However, if the authors even make assumptions about what the described change in the structure may lead to, this will improve the article and may cause additional attention of readers.
Comment 6
Finally the main comment. I think, the conclusions are lacking in completeness. In order to make the article acceptable for publication in Materials, I recommend two ways to authors choose. The first is to add the recommendations for practical application, for obtaining real objects for industry with a given structure. Clear recommendations based on the described results can be very valuable for production. The second way is to strengthen the paper from a fundamental point of view. The authors noticed and described interesting and important features, but did not clearly explain their nature, except that they are due to the influence of temperature. It is necessary to explain the mechanism of changes in the microstructure in such a way that the results and conclusions can be extrapolated to a wider range of materials. Probably there are previously described similar mechanisms of microstructure change in the literature. In this case, it is necessary to describe how the obtained results correlate with existing models. You can see a good example of such work here https://doi.org/10.1149/2.1001904jes .
Author Response
We are very much grateful to you for all the comments of great value. Here are our answers to your comments point-by-point. The corresponding revisions are highlighted with yellow color in our revised manuscript. Please also note that there were substantial revisions of expressions over all the manuscript irrespective of your specific comments.
For your overall positive evaluation on our manuscript, we would like to express our sincere thanks, and we wish that our revised manuscript can satisfy all your comments faithfully.
Concerning comment 1, we revised the introduction part. The important revisions according to this comment are highlighted in the revised manuscript. You can find them in the following locations.
> page 3, lines 9-11, 14-16
> page 3 line 23 – page 4 line 2
> page 20, lines 22-23: a new reference was added.
In addition, the justification for the material choice was also addressed in the beginning of the materials and methods section of the original manuscript, but might be too specific. We hope that our revision on the introduction can be more effective with this part.
For comment 2, we would like to simply answer here. Indeed in the original Fig. 9 (revised Fig. 7), the differences in the average grain diameter were within only a few microns. This implies that the microstructural inhomogeneity is not big in the AM, the inherent strength of this manufacturing technology which we stated in the introduction part. In spite of the small differences, there were clear trends observed in the original Fig. 9 (revised Fig. 7) which were consistently correlated with the process condition and the local thermal cycles in the discussion part. In our microstructures, the retained austenite phases are sensitive to cutting and polishing, they can transform to martensite in those processes without enough care. However, we were extremely careful about the instability of the retained austenite during our sample preparation, these were well reflected in our choices for the sample preparation methods in the materials and methods section, i.e. chemo-mechanical polishing or electropolishing to remove the distortion due to cutting, grinding and polishing. In spite of the thermodynamic instability of the retained austenite, the grain structure of the prior austenite which was determined at considerably high temperatures is hard to be distorted at room temperature, and the retained austenite is a minor phase in our case, thus hardly influences the reconstructed austenitic microstructures.
Thanks to comment 3, the original Fig. 6 and 9 (revised Fig. 6 and 7) have error bars now. We revised the figure captions correspondingly, too.
Concerning comment 4, we thought that you are right and decided to remove the original Fig. 7 which was not efficient in the characterization of the grain size distribution. In the original manuscript, we compared the two types of histograms (number frequency vs. area fraction) from the same data. However, from your comment, we thought that this was not essential for the purpose of our paper, but can deteriorate readability. Since the original Fig. 9 (revised Fig. 7) was a summarization of the original Fig. 7 and 8, we even decided to use only the former for clearness and compactness. You can find the corresponding revisions in the following locations.
> page 9, lines 10-19
> Fig. 7
Concerning comment 5, we greatly appreciate this kind guidance with the excellent literatures. But we apologize at first that we could not cite those articles because of the substantial differences in the specific fields of research. As you recommended, an article of materials science would be better with the full combination of composition, structure and property. We know that our current paper omitted the last one which should be the most important for practical applications. However, in the introduction part, we stated that the aim of our current manuscript was to provide a useful method of microstructural characterization which was not well established for the AM objects of martensitic steels. Thus, the correlation with properties is our next step planned. For current materials, we have difficulties of too small stock of powder and the very small sample dimensions, thus we only have one-point measurements of hardness currently, i.e. 53.5 HRC of C50 and 57.6 HRC of C2k. We thought the hardness can be correlated with the microstructural features described in the current manuscript partly, however, many other features such as porosity (described in the authors prior work on the same material [22]) should have a significant contribution. Thus, a full discussion with hardness only needs too much extension of the manuscript, which is far beyond the current aim and scope. Additionally, the measured hardness was not reliable enough (1-point measurements). Currently, we are preparing new materials and designing experiments to get a more classical article to be more helpful, such as the ones that you suggested. We ask for your kind understanding for our current deficiencies. But we think that our current manuscript is faithful at least to the objective which is stated in the introduction. We hope that you acknowledge it.
Lastly, you suggested two options for the revision of the conclusions. We followed the first one, and the revisions can be found in the following locations.
> page 16, lines 6-13
The second option would be more appropriate also concerning comment 5 and the article that you kindly suggested to read. However, there are currently too many things unrevealed in the selective melting of metals. More specifically, we are still unsure of the pattern of the austenitic grain growth prior to transformation because we can only know the final shape and dimension. Therefore, we could not provide a clear picture of the mechanism in the current manuscript. Again, we ask for your kind understanding and hope that you wait for our next-step work which should be more complete.
The above letter can be also accessed by the attached file (Reviewer2.docx)

Round 2
Reviewer 2 Report
Paper can be accepted after editor's evaluation.